# PARAMETER-EFFICIENT FINE-TUNING WITH CIRCULANT AND DIAGONAL VECTORS

## ABSTRACT

Foundation models have achieved tremendous success in different domains. However, their huge computation and storage complexity make these models difficult to fine-tune and also less applicable in practice. Recent study shows training in Fourier domain can be an effective fine-tuning method in terms of both model performance and number of training parameters. In this work, we propose to further reduce the complexity by the factorization through the product of interleaved circulant and diagonal matrices. Our method avoids the construction of weight change matrix and utilizes 1D fast Fourier transform (FFT) instead of 2D FFT. Experimental results show that our method achieves similar or better performance across various tasks with much less floating-point operations (FLOPs) and the number of trainable parameters. Compared with other Fourier domain based fine-tuning methods, the FLOPs of the RoBERTa base model achieves $33.1\times$ reduction, while the ViT base model achieves $7.76\times$ reduction. For number of trainable parameters of the RoBERTa base model, our method is $5.33\times$ smaller than LoRA, and for ViT base model ours can be $10.7\times$ smaller.

## 1 INTRODUCTION

Large foundation models (LFMs) are widely utilized in various fields, including natural language processing (Paaß & Giesselbach (2023)), image recognition and generation (Li et al. (2024a)), medical diagnosis (Li et al. (2024b)), and autonomous driving (Chen et al. (2024)). Devlin (2018) have proposed the bidirectional transformer architecture that understands input data from left to right and right to left. It is trained to predict missing words given input context, and it has served as a foundation model that can be fine-tuned for many downstream tasks. Following the transformer architecture, generative pre-trained transformer (GPT) model by Radford & Narasimhan (2018) handles input data from left to right following a sequential prediction order. This mechanism turns out successful in many generation tasks such as text summary, question answering, etc.

Although LFMs learn extensive general knowledge during the pre-training phase, they still require extra adjustments in downstream applications to effectively fullfill the task. Fine-tuning is a typical approach to continue learning on given downstream data and update from pre-trained model parameters. While fine-tuning significantly reduces computational costs compared to training from scratch, existing fine-tuning methods still suffer from the huge complexity of LFMs. As the original model parameters are still kept and maintained during fine-tuning stage, this leaves limited space for the development of fine-tuning methods.

To address the challenge of fine-tuning LFMs, Hu et al. (2021) have proposed low-rank adaptation (LoRA). This method is an efficient fine-tuning approach designed for LFMs, reducing the number of parameters required during fine-tuning by introducing low-rank matrices. The essential idea is assuming the weight change matrix with low rank structure, expressing it as the product of two low rank matrices and only training these two smaller matrices while keeping the original weights frozen. FourierFT proposed by Gao et al. (2024) assumes a sparse structure in fourier domain of the weight matrix updates $\mathbf{\Delta W}$. Although FourierFT reduces the number of training parameters, the computational and storage requirements of the model remain very high, particularly when dealing with LFMs. The two-dimensional Fourier transform used to restore $\mathbf{\Delta W}$ contributes to most of its computation and storage complexity. As a result, the fine-tuning model continues to necessitate

high-performance hardware support, including substantial GPU resources and memory, which may be challenging to achieve in practical applications.

Huhtanen & Perämäki (2015) has demonstrated that a general complex matrix $\mathbf{X} \in \mathbb{C}^{n \times n}$ can be factorized into the product of multiple circulant matrices and diagonal matrices, with total number of factors not exceeding $2n - 1$. This decomposition method offers several advantages, particularly in terms of computational efficiency and storage optimization. The computation and storage of diagonal matrices can be efficiently managed using vector representations. Moreover, circulant matrices possess a unique structure that allows them to be diagonalized using the fast Fourier transform (FFT), significantly reducing the complexity of matrix operations and accelerating computation speed.

Inspired by previous works, we propose circulant and diagonal vector based fine-tuning (CDVFT), which is also a Fourier domain based method. Our method represents the weight change matrix $\mathbf{\Delta W}$ with the product of interleaved circulant and diagonal matrices. This factorization simplifies the matrix calculation process and reduces storage requirements. Due to the unique properties of matrix product for circulant and diagonal matrices, the quadratic computation complexity now becomes loglinear. Different from FourierFT based on 2D FFT, our fine-tuning process avoids the restoration of the weight change $\mathbf{\Delta W}$ and only takes 1D FFT operations. As a result, CDVFT can achieve efficient storage and computation at the same time. We summarize our main contributions as following:

- We introduce CDVFT method that represents $\mathbf{\Delta W}$ using the product of interleaved circulant and diagonal matrices. These matrices have linear storage complexity as each of them can be determined by a single weight vector. In practice, we find only using a few circulant and diagonal matrix is sufficient to perform fine-tuning.

- CDVFT avoids the restoration of weight change matrix and has loglinear computation complexity. The circulant matrix vector product can be transformed into 1D FFT, and diagonal matrix vector product is linear in nature. Thus, the overall computation complexity becomes loglinear.

- We evaluate our method on natural language understanding, instruction adjustment, and image classification. Experimental results show that our method achieves similar or even better results in terms of moder performance, number of training parameters and FLOPs. For example, for the ViT base model, our method results in $7.76\times$ FLOPs reduction compared to FourierFT and $10.7\times$ trainable parameters saving compared to LoRA, while resulting in similar or even better accuracy.

## 2 RELATED WORKS

Fine-tuning LFMs is a challenging problem due to the large model size and computation requirement. Although training LFMs from scratch is performed on cloud platforms like LLaMA model by Touvron et al. (2023), fine-tuning is often limited to a specific task and a low-cost computing environment. Besides, fine-tuning runs on a much smaller dataset than the pre-training dataset for LFMs. Thus, fine-tuning process is expected to be cost-effective. The overall complexity should be small and affordable in practice.

Full fine-tuning is a classical approach training and updating all model parameters at the same time. However, it is difficult to perform full fine-tuning on LFMs given the huge computation and storage requirement. Brown et al. (2020) find LFMs are able to generalize to new tasks with few-shot demonstrations as prompt, thereby saving the effort of training on parameters. Li & Liang (2021) argue that adding few-shot demonstrations is bounded by the input length constraint of current LFMs. Instead, they propose the prefix tuning method to train a parameter vector and prepend to input, which is expected to work as prompt in unlimited length.

Updating all model parameters is not desirable in practice, since each task needs to maintain a model. Houlsby et al. (2019) propose the adapter method, where task dependent parameters are inserted to LFMs. Fine-tuning process only updates those new parameters, thereby each task effectively sharing pre-trained LFM parameters. Mahabadi et al. (2021) further reduce the task dependent parameters amount by grouping adapters into a hyper network model such that the network can produce task specific parameters on the fly. Sung et al. (2022) discover backpropagation process through LFMs

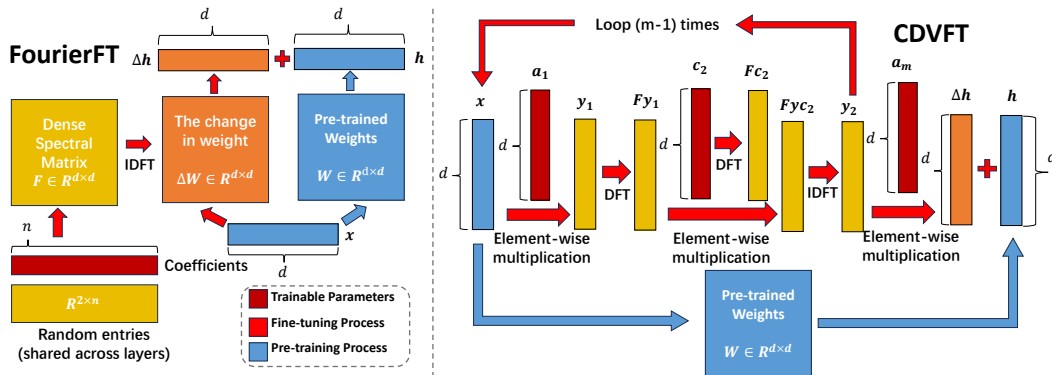

Figure 1: **Overview of FourierFT (left) and our CDVFT (right).** In FourierFT, one coefficient vector $\mathbf{c} \in \mathbb{R}^n$ is trained, and it is used to construct the weight change $\mathbf{\Delta W}$ through 2D FFT operation. In contrast, our CDVFT avoids the construction of $\mathbf{\Delta W}$, where matrix vector products are transformed into vector operations, i.e., element-wise product and 1D FFT, significantly reducing computation complexity and memory requirement. In practice, we find $m = 1$ (no loops required) can effectively fine-tune the model, where there are two diagonal matrices and one circulant matrix.

takes a lot of memory and propose a ladder style adapter design that significantly saves memory consumption. Given that adapters bring in extra inference latency due to their new parameters, Lei et al. (2023) believe different tasks have different needs for the shared LFM architecture. They decide to learn to skip computations in LFM for different adapters, resulting in a faster inference speed.

It can be noticed that adapter adds task dependent parameters and incurs inference delay. There are also studies working on mergeable adapters so that after fine-tuning they can be merged into LFM architecture without adding inference latency. The essential idea is setting adapter parameters in the same shape as LFM pre-trained parameters, and fine-tuning learns the change of weight parameters, i.e., $\mathbf{\Delta W}$. Hu et al. (2021) develop LoRA technique that enforces low rank structure into the weight change matrix. Given that LoRA rank can be different for different tasks, Zhang et al. (2023) decide to learn the rank setting by modifying singular values based on importance score function. Instead of directly learning on $\mathbf{\Delta W}$, Gao et al. (2024) propose FourierFT to learn sparse parameters in fourier domain and reconstruct the weight difference using 2D FFT operation. It turns out this method requires much less number of parameters, but its reconstruction needs more memory.

Following the parameter efficient fine-tuning (PEFT) discovery in fourier domain, it is important to look for a method involving matrix and efficient FFT operation. Circulant matrix is related to 1D FFT since circulant matrix vector product can be executed using 1D FFT to accelerate. There are some studies applying circulant matrix to compress neural networks, such as circulant convolution neural network by Cheng et al. (2015) and circulant long short-memory by Wang et al. (2018). However, these works are lack of flexibility on increasing parameter amount and theoretical guarantee on dense matrix approximation. Huhtanen & Perämäki (2015) has demonstrated that a general complex matrix $\mathbf{X} \in \mathbb{C}^{n \times n}$ can be expressed as the product of interleaved circulant and diagonal matrices, with the number of factors not exceeding $2n - 1$:

$$\mathbf{X} = \mathbf{A}_{2n-1} \times \mathbf{C}_{2n-2} \times \ldots \times \mathbf{C}_{2j} \times \mathbf{A}_{2j-1} \times \ldots \times \mathbf{A}_3 \times \mathbf{C}_2 \times \mathbf{A}_1 \times \mathbf{x} \tag{1}$$

where for $j \in \{1, \ldots, n\}$, $\mathbf{A}_{2j-1}$ and $\mathbf{C}_{2j}$ are diagonal and circulant matrices, respectively. Thus, this decomposition theoretically can approximate any dense matrix, and it also enables control on parameter amount by setting number of factors.

## 3 METHOD

In this section, we introduce circulant and diagonal vector based fine-tuning (CDVFT) method, which is a mergeable adapter design similar to FourierFT. After fine-tuning, our trained circulant and

diagonal vectors can be used to build circulant and diagonal matrices, which are further combined to reconstruct $\mathbf{\Delta W}$ and merged into LFMs. However, most importantly, CDVFT does not need to recover $\mathbf{\Delta W}$ during real fine-tuning process, since the reconstruction results in high computation and storage complexity. Instead, our method takes advantage of the fast matrix multiplication algorithm from circulant and diagonal matrices involving 1D FFT and element-wise product to achieve the goal.

The overall computation flow is illustrated in Fig.1. It can be seen that CDVFT only takes vector operations at each step, thereby significantly reducing the computation and storage complexity. Specifically, according to the findings by Huhtanen & Perämäki (2015) and the unique properties of circulant and diagonal matrix operations, CDVFT first initializes corresponding vectors to represent these matrices. It then directly performs multiple element-wise multiplication and 1D FFT on input $\mathbf{x}$. Finally, it yields the output $\mathbf{\Delta h}$, which can be added to the output $\mathbf{h}$ from the original weight matrix $\mathbf{W}$.

## 3.1 Forward Step

Let $\mathbf{x} \in \mathbb{R}^{d \times 1}$ be an input column vector. Assume weight change matrix $\mathbf{\Delta W} \in \mathbb{R}^{d \times d}$ that can be decomposed into $2m-1$ factors with $m \leq d$. Thus, there are $m$ diagonal matrices and $m-1$ circulant matrices. For $j \in \{1, 2, \ldots, m\}$, each diagonal matrix is defined by a vector $\mathbf{a}_{2j-1} \in \mathbb{R}^{d \times 1}$, and each circulant matrix is defined by a vector $\mathbf{c}_{2j} \in \mathbb{R}^{d \times 1}$. More specifically, they can be expressed as following:

$$
diag(\mathbf{a}_{2j-1}) = \begin{bmatrix} \mathbf{a}_{2j-1}^1 & 0 & \ldots & 0 \\ 0 & \ddots & \ddots & \vdots \\ \vdots & \ddots & \ddots & 0 \\ 0 & \ldots & 0 & \mathbf{a}_{2j-1}^d \end{bmatrix}, \quad circ(\mathbf{a}_{2j}) = \begin{bmatrix} \mathbf{a}_{2j}^1 & \mathbf{a}_{2j}^d & \ldots & \mathbf{a}_{2j}^2 \\ \mathbf{a}_{2j}^2 & \ddots & \ddots & \vdots \\ \vdots & \ddots & \ddots & \mathbf{a}_{2j}^d \\ \mathbf{a}_{2j}^d & \ldots & \mathbf{a}_{2j}^2 & \mathbf{a}_{2j}^1 \end{bmatrix}, \quad (2)
$$

where $diag(\cdot)$ and $circ(\cdot)$ construct a diagonal matrix and circulant matrix, respectively. Therefore, the weight change matrix can be written as:

$$
\begin{aligned}
\mathbf{\Delta W} &= \mathbf{A}_{2m-1} \times \mathbf{C}_{2m-2} \times \mathbf{A}_{2m-3} \times \cdots \times \mathbf{A}_1 \\
&= diag(\mathbf{a}_{2m-1}) \times circ(\mathbf{a}_{2m-2}) \times diag(\mathbf{a}_{2m-3}) \cdots \times diag(\mathbf{a}_1),
\end{aligned} \quad (3)
$$

where $\times$ is the inner product operation. The end-to-end computation flow then becomes:

$$
\mathbf{h}' = \mathbf{h} + \mathbf{\Delta h} = \mathbf{W} \times \mathbf{x} + \alpha \times \mathbf{\Delta W} \times \mathbf{x}, \quad (4)
$$

where $\alpha$ is a hyper-parameter scalar as in LoRA (Hu et al., 2021), $\mathbf{W} \in \mathbb{R}^{d \times d}$ is the pre-trained weight matrix in given LFM and $\mathbf{h}'$ is the new output after adding our CDVFT adapters. This can also be seen in Fig. 1.

We perform the computation from rightmost to leftmost, thereby avoiding the reconstruction of $\mathbf{\Delta W}$ during fine-tuning process. Let $\mathbf{y} \in \mathbb{R}^{d \times 1}$ represent the intermediate calculation result from matrix vector multiplications. Thus, $\mathbf{y}_{2j-1}$ is the result from diagonal matrix vector multiplication, and $\mathbf{y}_{2j}$ is the result from circulant matrix vector multiplication. Note that diagonal matrix vector product is equivalent to element wise product of $\mathbf{a}_{2j-1}$ and input vector:

$$
\mathbf{\Delta W} \times \mathbf{x} = \mathbf{A}_{2m-1} \times \ldots \times \overbrace{\mathbf{C}_{2j} \times \underbrace{\mathbf{A}_{2j-1} \times \ldots \times \mathbf{A}_3 \times \mathbf{C}_2 \times \mathbf{A}_1 \times \mathbf{x}}_{\mathbf{y}_{2j-1}}}^{\mathbf{y}_{2j}}, \quad (5)
$$

$$
\mathbf{y}_{2j} = \mathbf{C}_{2j} \times \mathbf{y}_{2j-1}, \quad \mathbf{y}_0 = \mathbf{x}, \quad (6)
$$

$$
\mathbf{y}_{2j-1} = \mathbf{A}_{2j-1} \times \mathbf{y}_{2j-2} = \mathbf{a}_{2j-1} \odot \mathbf{y}_{2j-2}, \quad (7)
$$

where $\odot$ means the element-wise product. The circulant matrix vector product can be transformed into 1D FFT operations:

$$
\mathbf{Fy}_{2j-1}^p = \sum_{q=0}^{d-1} \mathbf{y}_{2j-1}^q e^{-i2\pi \frac{p}{d} q}, \quad \mathbf{Fc}_{2j}^p = \sum_{q=0}^{d-1} \mathbf{c}_{2j}^q e^{-i2\pi \frac{p}{d} q},
$$

$$
\mathbf{Fyc}_{2j} = \mathbf{Fy}_{2j-1} \odot \mathbf{Fc}_{2j}, \quad \mathbf{y}_{2j}^p = \frac{1}{d} \sum_{q=0}^{d-1} \mathbf{Fyc}_{2j}^q e^{i2\pi \frac{p}{d} q}, \quad (8)
$$

Table 1: Trainable parameters amount and storage cost of different fine-tuning methods. For all methods and foundation models, only the query and value weight matrices in model attention architecture are fine-tuned. $r$ is the rank setting for LoRA, $n$ is number of parameters in Fourier domain set in FourierFT, and $m$ is number of factors of our CDVFT. Improvements with respect to LoRA is also reported. For example, in terms of ViT-Base, the parameter amount improvement of FourierFT over LoRA is around 8.19, and ours is 10.7, which means our CDVFT uses less number of parameters than FourierFT.

| | Base Models | RoBERTa-Base | ViT-Base |
|---|---|---|---|
| LoRA | r | 8 | 16 |
| | # Trainable Parameters | 295K(1.00×) | 590K(1.00×) |
| | Required Bytes | 1.13MB | 2.25MB |
| FourierFT | n | 1000 | 3000 |
| | # Trainable Parameters | **24.0K**(12.3×) | 72.0K(8.19×) |
| | Required Bytes | 94KB | 281KB |
| Ours | m | 2 | 2 |
| | # Trainable Parameters | 55.3K(5.33×) | **55.3K**(10.7×) |
| | Required Bytes | 217KB | 217KB |

where $e^{i2\pi\frac{p}{d}q}$ is the constant term in the Fourier transform, $i$ indicates the imaginary unit, and $p$ is the frequency index of the transform. We use letter $\mathbf{F}$ to indicate vectors in fourier domain. $\mathbf{Fy}_{2j-1}$ and $\mathbf{Fc}_{2j}$ represent the Fourier transform results of $\mathbf{y}_{2j-1}$ and the circulant matrix vector $\mathbf{c}_{2j}$, respectively. $\mathbf{Fyc}_{2j}$ is the result of element wise multiplication of $\mathbf{Fy}_{2j-1}$ and $\mathbf{Fc}_{2j}$. In consequence, $\mathbf{y}_{2j}$ is the result of inverse fast Fourier transform (IFFT) of $\mathbf{Fyc}_{2j}$.

## 3.2 BACKWARD STEP

Following current deep learning design, we provide the gradient calculation with respect to $\mathbf{a}_{2j-1}$ and $\mathbf{c}_{2j}$ for all $j$. Denote the objective function (i.e., loss function) as $\mathcal{L}(\cdot)$. The backpropagation follows the chain rule, and we can get:

$$\frac{\partial\mathcal{L}}{\partial\mathbf{a}_{2j-1}} = \frac{\partial\mathcal{L}}{\partial\mathbf{y}_{2j-1}}\frac{\partial\mathbf{y}_{2j-1}}{\partial\mathbf{a}_{2j-1}} = \frac{\partial\mathcal{L}}{\partial\mathbf{y}_{2j-1}}\odot\mathbf{y}_{2j-2}. \tag{9}$$

The backpropagation through the circulant matrix consists of derivatives of one-dimensional Fourier transform, which is easier to derive with explicit expression of FFT as shown in Eq. (8):

$$\frac{\partial\mathcal{L}}{\partial\mathbf{Fyc}_{2j}^q} = \frac{\partial\mathcal{L}}{\partial\mathbf{y}_{2j}}\frac{\partial\mathbf{y}_{2j}}{\partial\mathbf{Fyc}_{2j}^q} = \frac{1}{d}\sum_{p=0}^{d-1}\frac{\partial\mathcal{L}}{\partial\mathbf{y}_{2j}^p}e^{i2\pi\frac{p}{d}q}, \tag{10}$$

$$\frac{\partial\mathcal{L}}{\partial\mathbf{Fc}_{2j}} = \frac{\partial\mathcal{L}}{\partial\mathbf{Fyc}_{2j}}\odot\mathbf{Fy}_{2j-1}, \quad \frac{\partial\mathcal{L}}{\partial\mathbf{Fy}_{2j-1}} = \frac{\partial\mathcal{L}}{\partial\mathbf{Fyc}_{2j}}\odot\mathbf{Fc}_{2j}, \tag{11}$$

$$\frac{\partial\mathcal{L}}{\partial\mathbf{c}_{2j}^q} = \frac{\partial\mathcal{L}}{\partial\mathbf{Fc}_{2j}}\frac{\partial\mathbf{Fc}_{2j}}{\partial\mathbf{c}_{2j}^q} = \sum_{p=0}^{d-1}\frac{\partial\mathcal{L}}{\partial\mathbf{Fc}_{2j}}e^{-i2\pi\frac{p}{n}q}, \tag{12}$$

$$\frac{\partial\mathcal{L}}{\partial\mathbf{y}_{2j-1}^q} = \frac{\partial\mathcal{L}}{\partial\mathbf{Fy}_{2j-1}}\frac{\partial\mathbf{Fy}_{2j-1}}{\partial\mathbf{y}_{2j-1}^q} = \sum_{p=0}^{d-1}\frac{\partial\mathcal{L}}{\partial\mathbf{Fy}_{2j-1}^q}e^{-i2\pi\frac{p}{n}q}, \tag{13}$$

where it can be noticed that the backpropagation also consists of element-wise product and 1D FFT operation on vectors.

## 3.3 COMPLEXITY ANALYSIS

Table 1 summarizes number of trainable parameters for LoRA, FourierFT, and CDVFT. Assume that the number of layers to be fine-tuned is $L_t$ and $\mathbf{\Delta W}\in\mathbb{R}^{d\times d}$. The number of parameters $\Theta$ to be trained for LoRA is given by $|\Theta|_{\text{LoRA}} = 2\times d\times L_t\times r$, where $|\cdot|$ means the cardinality. For FourierFT, let number of spectral coefficients be $n$, and the total number of trainable parameters is

Table 2: FLOPs of forward computation for different fine-tuning methods and foundation models. $n$ is parameter amount in Fourier domain, $r$ is rank setting, and $m$ is number of factors. Given the huge difference in FLOPs, FourierFT is chosen as baseline, and corresponding improvement is listed for each method.

| Base Models | FourierFT | | LoRA | | Ours | |
|---|---|---|---|---|---|---|
| | n | FLOPs | r | FLOPs | m | FLOPs |
| RoBERTa-Base | 1000 | 1.625G(1.00×) | 8 | **10.62M**(153.0×) | 2 | 49.03M(33.14×) |
| ViT-Base | 3000 | 4.159G(1.00×) | 16 | **0.232G**(17.93×) | 2 | 0.536G(7.759×) |

$|\Theta|_{\text{FourierFT}} = n \times L_t$. For CDVFT, assuming that the total number of circulant matrices and diagonal matrices is $2m - 1$, the total number of trainable parameters is $|\Theta|_{\text{CDVFT}} = (2m - 1) \times d \times L_t$. In the case of ViT base model, $d = 768$. There are $L_t = 24$ attention layers to be fine-tuned, and it should be noted that fine-tuning only runs on the query and value weight matrices. Corresponding parameter counts are as follows: for LoRA, when $r = 8$, $|\Theta|_{\text{LoRA}} = 294912$; for FourierFT, when $n = 3000$, $|\Theta|_{\text{FourierFT}} = 72000$; and for CDVFT, when $m = 2$, $|\Theta|_{\text{CDVFT}} = 55296$. Table 1 shows that Fourier domain based method, i.e., FourierFT and ours, require much less number of parameters than LoRA. It can be noted that our CDVFT uses same number of parameters for RoBERTa and ViT models because both $d$ and number of fine-tuned layers are the same. The improvements with respect to LoRA method is also reported for all methods. Compared with FourierFT, our method achieves comparable parameter reduction over LoRA.

Further analysis of the computational complexity of FourierFT and CDVFT is shown in Fig.2. It is important to note that computation complexity of FourierFT is independent of its parameter amount $n$ since it always use 2D FFT to reconstruct $\Delta \mathbf{W}$. The complexity of our CDVFT is related to total number of circulant matrices and diagonal matrices, i.e., $2m - 1$. In this paper, we find $m = 2$ is sufficient to perform effective fine-tuning in practice. The computational complexity of CDVFT is smaller than FourierFT. The main reason for the complexity difference is that in FourierFT, the computational complexity of the 2D FFT for computing $\Delta \mathbf{W}$ is $O(d^2 log(d^2))$. In CDVFT, the complexity brought by element-wise product and 1D FFT is $O(mdlog(d))$, which significantly reduces the computational complexity while keeping similar number of training parameters. To compare different fine-tuning methods, we present the corresponding FLOPs comparison in Table 2. The improvement with

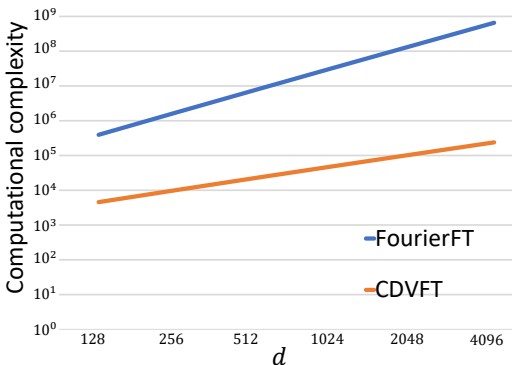

Figure 2: The computational complexity comparison of Fourier domain based method. The horizontal axis represents the size of $d$. Note that the computational complexity of FourierFT is independent of its parameter amount $n$. For our CDVFT, $m = 2$, so there are 3 matrix factors.

respect to FourierFT is reported besides FLOPs amount. It can be noted that our method results in different FLOPs while using the same number of trainable parameters as shown in Table 1. This is caused by different sequence length of attention architecture in RoBERTa and ViT. Overall, FourierFT needs much larger FLOPs than both LoRA and our CDVFT.

## 4 EXPERIMENTS

In this section, we evaluate our CDVFT method across different domains, i.e., natural language understanding (NLU) and computer vision (CV): (1) fine-tune the RoBERTa model (Liu et al., 2019) on the General Language Understanding Evaluation (GLUE) dataset (Wang et al., 2019); (2) fine-tune the vision transformer model (Dosovitskiy et al., 2021) for various image classification tasks across different domains. Our proposed CDVFT is also compared with LoRA (Hu et al., 2021) and FourierFT (Gao et al., 2024). LoRA is a widely adopted LFM fine-tuning method due to its ease of

Table 3: Hyperparameter setup of CDVFT for the GLUE benchmark.

| Hyperparameter | CoLA | SST-2 | MRPC | STS-B | QNLI | RTE |
|---|---|---|---|---|---|---|
| Optimizer | AdamW | | | | | |
| LR Schedule | Linear | | | | | |
| Warmup Ratio | 0.06 | | | | | |
| m | 2 | | | | | |
| Epochs | 100 | 40 | 30 | 80 | 40 | 90 |
| Learning Rate(QV) | 1.2E-1 | 5E-2 | 4E-2 | 8E-2 | 1E-1 | 9E-2 |
| Learning Rate(Head) | 8E-3 | 6E-3 | 4E-2 | 9E-3 | 1E-3 | 1.1E-2 |
| Max Seq. Len | 512 | 512 | 512 | 512 | 512 | 512 |
| Scaling value | 5E-5 | 5E-4 | 5E-4 | 5E-5 | 1E-4 | 1E-4 |
| Batch Size | 32 | 32 | 32 | 32 | 32 | 32 |

Table 4: The performance of LoRA, FourierFT and our CDVFT methods is reported by fine-tuning the RoBERTa base model on 6 datasets of the GLUE benchmark. The experiments report Matthew correlation coefficient (MCC) for CoLA, Pearson correlation coefficient (PCC) for STS-B, and accuracy (Acc.) for all remaining tasks. Following (Gao et al., 2024), we also report the median result out of 5 runs, each with a different random seed. The best result for each dataset is highlighted in bold. Higher metric value means better model performance for all datasets.

| Model & Method | $\text{RoB}_{base}(LoRA)$ | $\text{RoB}_{base}(FourierFT)$ | $\text{RoB}_{base}(Ours)$ |
|---|---|---|---|
| CoLA (MCC) | $63.4_{\pm 1.2}$ | $63.8_{\pm 1.6}$ | $\mathbf{64.5}_{\pm 1.2}$ |
| SST-2 (Acc.) | $\mathbf{95.1}_{\pm 0.2}$ | $94.2_{\pm 0.3}$ | $94.4_{\pm 0.5}$ |
| MRPC (Acc.) | $89.7_{\pm 0.7}$ | $90.0_{\pm 0.8}$ | $\mathbf{90.2}_{\pm 0.3}$ |
| STS-B (PCC) | $\mathbf{91.5}_{\pm 0.2}$ | $90.8_{\pm 0.2}$ | $90.5_{\pm 0.2}$ |
| QNLI (Acc.) | $\mathbf{93.3}_{\pm 0.3}$ | $92.2_{\pm 0.1}$ | $92.2_{\pm 0.2}$ |
| RTE (Acc.) | $78.4_{\pm 0.8}$ | $\mathbf{79.1}_{\pm 0.5}$ | $78.7_{\pm 1.2}$ |
| Avg. | $\mathbf{85.2}$ | $85.0$ | $85.1$ |

implementation and effective adaptation. FourierFT serves as a baseline for FFT based fine-tuning method that requires much less number of parameters than LoRA.

## 4.1 NATURAL LANGUAGE UNDERSTANDING

**Models and Datasets.** We evaluate CDVFT on the GLUE benchmark dataset, which consists of a diverse range of NLP tasks, each representing a specific type of language understanding task. These tasks include question answering, sentiment analysis, textual entailment, etc. Following the experiment setting as in (Gao et al., 2024), fine-tuning process runs on following tasks: CoLA, Corpus of Linguistic Acceptability (Warstadt et al., 2019), which determines whether sentences adhere to grammatical rules; SST-2, Stanford Sentiment Treebank (Socher et al., 2013), which classifies the sentiment of sentences as positive or negative; MRPC, Microsoft Research Paraphrase Corpus (Dolan & Brockett, 2005), which assesses whether two sentences convey the same meaning; STS-B, Semantic Textual Similarity Benchmark (Cer et al., 2017), which measures the semantic similarity score between sentence pairs; QNLI, Question Natural Language Inference (Rajpurkar, 2016), which evaluates whether the second sentence correctly answers the question posed by the first; and RTE, Recognizing Textual Entailment (Dagan et al., 2005), which identifies whether there is an entailment relationship between sentence pairs, functioning as a binary classification task. RoBERTa base model (Liu et al., 2019) is a transformer based foundation model, which is widely used in natural language processing. It improves over existing under-trained BERT model (Devlin, 2018) while preserving the powerful attention mechanism. Thus, it is selected to serve as the foundation model for GLUE dataset.

**Implementation Details.** Our CDVFT uses a total of 3 factor matrices, i.e., $m = 2$. The detailed hyperparameters are shown in Table 3. It should be noted that only query and value weights in each transformer block are finetuned, which is also applied to FourierFT and LoRA as in (Gao et al., 2024). All implementations are in PyTorch (Paszke et al., 2019). It can be seen that the optimizer is AdamW (Loshchilov, 2017). For each dataset, there are different learning rates for foundation

Table 5: Hyperparameter setup for image classification of CDVFT.

| Hyperparameter | OxfordPets | StanfordCars | CIFAR10 | CIFAR100 | DTD | EuroSAT | FGVC | RESISC45 |
|---|---|---|---|---|---|---|---|---|
| Optimizer | | | | AdamW | | | | |
| LR Schedule | | | | Linear | | | | |
| Warmup Ratio | | | | 0.06 | | | | |
| m | | | | 2 | | | | |
| Epochs | | | | 10 | | | | |
| Learning Rate (QV) | 3E-1 | 3E-1 | 3E-2 | 2E-1 | 3E-1 | 2E-1 | 3E-1 | 3E-1 |
| Learning Rate (Head) | 1E-3 | 1E-3 | 1E-3 | 7E-4 | 1E-3 | 8E-4 | 1E-3 | 1E-3 |
| Weight Decay | 8E-4 | 4E-5 | 9E-5 | 1E-4 | 7E-5 | 3E-4 | 7E-5 | 3E-4 |
| Scaling value | 1E-2 | 5E-3 | 1E-2 | 1.5E-3 | 5E-3 | 5E-2 | 1E-2 | 5E-3 |
| Batch Size | 50 | 50 | 50 | 50 | 50 | 50 | 50 | 50 |

Table 6: Fine-tuning results of the ViT Base model on different image classification datasets. The experiments report the accuracy (%) after 10 epochs.

| Model & Method | $ViT_{base}(LoRA)$ | $ViT_{base}(FourierFT)$ | $ViT_{base}(Ours)$ |
|---|---|---|---|
| OxfordPets | $93.19_{\pm 0.36}$ | $\mathbf{93.21}_{\pm 0.26}$ | $92.62_{\pm 0.37}$ |
| StanfordCars | $57.40_{\pm 0.66}$ | $57.14_{\pm 0.31}$ | $\mathbf{57.40}_{\pm 0.55}$ |
| CIFAR10 | $\mathbf{98.78}_{\pm 0.05}$ | $98.58_{\pm 0.07}$ | $98.61_{\pm 0.09}$ |
| CIFAR100 | $\mathbf{92.02}_{\pm 0.12}$ | $91.20_{\pm 0.14}$ | $91.11_{\pm 0.12}$ |
| DTD | $88.16_{\pm 0.91}$ | $86.19_{\pm 1.05}$ | $\mathbf{88.75}_{\pm 0.78}$ |
| EuroSAT | $98.44_{\pm 0.15}$ | $\mathbf{98.71}_{\pm 0.08}$ | $98.56_{\pm 0.14}$ |
| FGVC | $\mathbf{36.74}_{\pm 1.31}$ | $36.38_{\pm 2.33}$ | $36.60_{\pm 0.73}$ |
| RESISC45 | $\mathbf{92.70}_{\pm 0.18}$ | $93.22_{\pm 0.18}$ | $92.04_{\pm 0.12}$ |
| Avg. | $\mathbf{82.18}$ | $81.83$ | $81.96$ |

model language heads, query and value weight matrices. The scaling value is the $\alpha$ as in Eq. (4). The batch size and maximum input sequence length is set the same for all datasets.

**Results.** Table 4 summarizes fine-tuning results of all methods. The median metric value with standard deviation is reported out of 5 runs of experiments for each fine-tuning method, where each run takes a different random seed. The best performance for each dataset is highlighted in bold. Overall, compared with LoRA and FourierFT, our CDVFT method achieves comparable or even better performance. Besides, according to Table 1 and Table 2, our CDVFT results in $5.33\times$ less number of trainable parameters than LoRA and $33.14\times$ less FLOPs than FourierFT while fine-tuning RoBERTa base model on GLUE dataset.

## 4.2 IMAGE CLASSIFICATION

**Models and Datasets.** The experiment evaluates the performance of our CDVFT method in image classification tasks, utilizing the Vision Transformer (ViT) by Dosovitskiy et al. (2021) as the foundation model. Following the setting in (Gao et al., 2024), we fine-tune on several challenging image classification datasets: OxfordPets (Parkhi et al., 2012) contains cats and dogs images in multiple breeds but with subtle difference; StanfordCars (Krause et al., 2013) has fine-grained categories of cars; Describable textures dataset (DTD) by Cimpoi et al. (2014) studies object textures categories; EuroSAT (Helber et al., 2019) collects geo-referenced satellite images for various land uses; RE-SISC45 Cheng et al. (2017) provides a diverse range of remote sensing images; FGVC-Aircraft (Maji et al., 2013) contains rigid and less deformable aircrafts images; CIFAR-10 and CIFAR-100 Krizhevsky et al. (2009) are classical datasets of tiny images in 10 and 100 categories, respectively.

**Implementation details.** We set $m = 2$ for fine-tuning ViT base model across all these datasets. Detailed hyperparameters are shown in Table 5. For all method, fine-tuning only runs the query and value weight matrices of ViT, which is the same as in (Gao et al., 2024). The learning rate is set differently for fine-tuning ViT heads and query and value weight matrices.

**Results.** Table 6 summarizes the results on eight image classification datasets by fine-tuning the ViT base model. It can be noticed that the StanfordCars, DTD and FGVC dataset metrics are about 10% higher than the numbers reported in (Gao et al., 2024), because their related dataset split random seed is unclear. For the purpose of fair comparison, we re-run these experiemnts for FourierFT and

LoRA and are able to observe similar performance increase. Along with results in Table 1 and Table 2, our CDVFT takes $10.7\times$ less number of parameters than LoRA and $7.76\times$ less number of FLOPs than FourierFT while achieving similar and sometimes better classification accuracy.

## 5 CONCLUSION

Motivated by the recent success in Fourier domain based fine-tuning method, this paper proposes the CDVFT method that also learns parameters in Fourier domain. In particular, our method results in both trainable parameters savings and FLOPs reduction when compared with existing methods. The downstream task performance of our fine-tuned model achieves similar performance and sometime even better results across both natural language understanding and computer vision applications. These results effectively demonstrate the promising potential of our method and also the Fourier domain based fine-tuning methods.

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

# A APPENDIX

We show the forward process of CDVFT in algorithm 1. First, $m$ is used to determine the number of diagonal and circulant matrices, and then the size of the diagonal and circulant vectors is determined by dimension $d$. Then, $\Delta\mathbf{h} = \mathbf{A}_{2m-1} \times \ldots \times \mathbf{C}_{2j} \times \mathbf{A}_{2j-1} \times \ldots \times \mathbf{A}_3 \times \mathbf{C}_2 \times \mathbf{A}_1 \times \mathbf{x}$ is completed through element wise multiplication and one-dimensional Fourier transform, and finally the final output change is obtained, which is combined with the original output to obtain the final output of the fine-tuning layer.

---

**Algorithm 1** PyTorch-style pseudocode for **CDVFT**

---

```
class CDVFT(nn.Module):
    def __init__(
        self,
        m: int = 2,
        alpha: float = 1e-4, # scaling
        d: int = 4096,
        base_layer: nn.Module # pre-trained layer
    ):
        # definitions
        self.m = m
        self.d = d
        self.scale = alpha
        self.base_layer = base_layer
        # diagonal matrices and circulant matrix initialization
        self.diags = nn.ParameterList([nn.Parameter(torch.randn(1, self.n
)) for _ in range(i)])
        self.circs = nn.ParameterList([nn.Parameter(torch.randn(1, self.n
)) for _ in range(i-1)])

    def forward(self, x: torch.Tensor):
        for i in range(len(self.diags)):
            # compute diagonal matrix multiplication (Eq.2)
            x = x * self.diags[i].unsqueeze(0)
            # compute circulant matrix multiplication (Eq.3)
            if i < len(circs):
                fd = torch.fft.fft(x,dim=2)
                fc = torch.fft.fft(self.circs[i],dim=1)
                fdc = fd * fc.unsqueeze(0)
                x = torch.fft.ifft(fdc,dim=2)
        #compute delta output and merge (Eq.4, 5)
        x = x.real * self.scale
        result = self.base_layer(x)
        result += x
        return result
```

---

