# OpenReview forum: "Parameter-Efficient Fine-Tuning with Circulant and Diagonal Vectors"
_ICLR.cc/2025/Conference — Submitted to ICLR 2025_

### Official Review · Reviewer_QbRu · 2024-10-27

**Soundness:** 2
**Presentation:** 2
**Contribution:** 1
**Rating:** 3
**Confidence:** 3

**Summary:**

This paper introduces a matrix factoring method, consisting of productions of circulant and diagonal matrices, to represent the parameter update matrix and uses limited factors to save the trainable parameters and 1-d fast FFT to elegantly reduce the computational cost.

**Strengths:**

Proposing to introduce a new matrix factorization method into the fine-tuning process is a novel and interesting method for saving the number of parameters and has potential exploration value. The paper is written very clearly and is easy to understand. Experiments have been conducted on two different tasks of NLP and CV, enhancing the method's reliability.

**Weaknesses:**

My main concern focuses on the experimental performance of this method. Although the proposed method requires fewer trainable parameters, it costs more in computation(Flops) and performs worse than LoRA. Although having fewer trainable parameters is good, the runtime memory and computational cost may be more important during fine-tuning. These parameters can be merged into the original parameter matrix during the final inference.
The benchmark methods chosen in the article are not the most advanced currently. It may be necessary to select some SOTA methods for comparison, especially with similar trainable parameter numbers and computational costs. Tests can be conducted on more datasets such as the MMLU dataset.


Typo in the appendix: the variable i should be m in the init function.


@article{hendryckstest2021,
  title={Measuring Massive Multitask Language Understanding},
  author={Dan Hendrycks and Collin Burns and Steven Basart and Andy Zou and Mantas Mazeika and Dawn Song and Jacob Steinhardt},
  journal={Proceedings of the International Conference on Learning Representations (ICLR)},
  year={2021}
}

**Questions:**

How to handle the case where W is not a square matrix?

How is m determined here? I am very curious about the improvement of the method's expressive power by increasing $m$. Especially when the performance is still not as good as the LoRA, appropriately increasing m to 4 should still be profitable compared to $r = 8$. And I am very concerned about the runtime memory usage of this method. Can the author compare it with other methods?

Could the author further explain why this method is Fourier domain-based method? Could the impact of truncating and preserving some factors on the overall representational ability be illustrated?

---

### Official Review · Reviewer_Xx7i · 2024-10-29

**Soundness:** 2
**Presentation:** 2
**Contribution:** 2
**Rating:** 5
**Confidence:** 4

**Summary:**

This paper introduces CDVFT, which facilitates parameter-efficient finetuning through 1D-FFT-based matrix decomposition. It employs trainable diagonal and circulant matrices to compute the weight change matrix. Experimental results show that CDVFT achieves comparable model performance with the baselines, while trading-off the number of trainable parameters and training FLOPs.

**Strengths:**

-	This paper draws inspiration from matrix decomposition using diagonal and circulant matrices, and integrates the existing idea into the parameter-efficient finetuning problem.
-	The paper includes a detailed analysis of the number of trainable parameters and training complexity.
-	Experimental results on various workloads with multiple runs validate the effectiveness of the proposed method.

**Weaknesses:**

-	The motivation for CDVFT is not convincing. It is unclear if balancing the number of trainable parameters (which affects memory requirements) with training FLOPs (which affects training time) is necessary. In the experiments presented, the number of trainable parameters and required memory are already negligible, making further reduction seem unnecessary.
-	The paper lacks a clear explanation for the selection of hyperparameters. While it claims “m=2” is sufficient, it does not address whether increasing m would improve model performance or how it balances model quality with efficiency. Furthermore, the rationale behind the hyperparameter choices for baseline methods is not provided. There is no analysis on r, n, and m, which could lead to different ranks (or information capacity) of trainable parameters, affecting model quality.
-	The experiment is insufficient. The paper compares only two baselines, with LoRA being outdated, overlooking many newer methods in the domain. Additionally, there is no ablation study on hyperparameters.

**Questions:**

-	Is it necessary, and when is it necessary to trade-off between the number of trainable parameters and training FLOPs?
-	In the experiments, the number of trainable parameters is already negligible. Is it necessary to further reduce the number of trainable parameters compared to LoRA?
-	How do the hyperparameters r, n, m influence the results? Since these methods involve compression of trainable parameters, how do these hyperparameters affect the rank of the parameters, and how do they further affect model quality? Conducting an ablation study with thorough analysis is essential.
-	The paper only discusses LoRA and 2D FFT methods. Given the numerous new parameter-efficient fine-tuning methods, it's important to include experiments or discussions on these approaches.
-	What is the specific motivation for choosing this decomposition method for parameter-efficient fine-tuning, given many decomposition methods available?
-	The presentation could be improved. For example, in Figure 1, the legend explains only some of the colors.

---

### Official Review · Reviewer_vn2t · 2024-10-29

**Soundness:** 3
**Presentation:** 2
**Contribution:** 3
**Rating:** 3
**Confidence:** 3

**Summary:**

This paper proposes a novel parameter-efficient fine-tuning technique called circulant and diagonal vector based fine-tuning (CDVFT), which is also a Fourier domain-based method. It uses the product of interleaved circulant and diagonal matrices (i.e., the factorization) as the adapter, claiming the quadratic computation complexity now becomes loglinear. The theoretical analysis illustrates the superiority in the computational and storage requirements. The authors conducted experiments regarding the fine-tuning of NLP and CV models.

**Strengths:**

1. The idea of using the product of interleaved circulant and diagonal matrices to represent the weight updates is interesting.
2. The theoretical analysis of computational complexity is good to illustrate the benefit of CDVFT in terms of FLOPs.
3. The technical details are sufficient and friendly to reproduce.

**Weaknesses:**

1. The experiments regarding NLP tasks are insufficient. Authors should consider experiments of fine-tuning the latest models such as Llama-3-8B. Moreover, authors should consider the evaluation of CDVFT on the generation tasks.
2. The improvement of computation complexity may not indicate the reduction of end-to-end fine-tuning costs. The factorization can bring exorbitant costs on the kernel launch which might offset the benefit of the computation complexity improvement.
3. The evaluation does not include the performance of full fine-tuning, which is necessary for suggesting the performance difference between your proposed method and a strong baseline.
4. The authors should include an experiment that illustrates how performance varies under different "m" (the loop times in CDVFT).

**Questions:**

I am curious about the end-to-end fine-tuning cost of CDVFT (and its comparison with others), and how it performs on the latest LLMs.

---

### Official Review · Reviewer_aS7T · 2024-11-02

**Soundness:** 2
**Presentation:** 2
**Contribution:** 3
**Rating:** 5
**Confidence:** 3

**Summary:**

This paper introduces CDVFT (Circular and Diagonal Vector-based Fine-tuning), a parameter-efficient fine-tuning technique for large foundation models (LFMs) in various tasks. Compared with traditional techniques using 2D FFT, CDVFT uses circular and diagonal matrix factorization and focuses on 1D FFT to reduce the computational cost and storage requirements of fine-tuning LFMs. Experimental results show that it is competitive in image classification and natural language processing applications with significantly reduced number of parameters and floating-point operations (FLOPs).

**Strengths:**

1. CDVFT is more efficient than LoRA and FourierFT by using matrix factorization and 1D FFT to effectively reduce the number of parameters and FLOPs.

2. CDVFT exploits circulant and diagonal matrix product properties to provide computational advantages, thereby significantly reducing FLOPs and complexity.

3. Extensive testing on RoBERTa and ViT models confirms that the proposed strategy performs as well or better than baseline techniques, and the results show that it requires less computational power.

**Weaknesses:**

1. Although the study achieved impressive empirical results, there is a lack of thorough theoretical analysis and evidence to confirm the advantages of the circulant-diagonal factorization method.

2. The reliance of the method on FFT-based operations may limit its generality as it is only applicable to LFMs with specific architectural characteristics.

3. The study did not investigate various hyperparameter combinations beyond the fixed setting, which may affect the robustness of the CDVFT results in various scenarios.

**Questions:**

1. Can more theoretical research improve the understanding of the model's efficacy, especially regarding the advantage of circulant diagonal matrix products?

2. Is the method limited to models for which FFT-based representations are practical, or is it applicable to architectures other than those using 1D FFTs?

3. How will CDVFT perform if the m parameter changes or new hardware limitations arise? Does every foundation model benefit from the 2-factor setting?

---

### Meta-Review · Area_Chair_p5QP · 2024-12-18

**Metareview:**

Reviewers mainly have these concerns:
- Lacking of thorough theoretical analysis and evidence to confirm the advantages of the circulant-diagonal factorization method.
- insufficient experimental settings.
- Missing some details about hyperparameter selection and comparision results.
All reviewers give the recommondation of rejection and the authors do not provide any response during and after rebuttal period. Thus, this paper should be rejected.

**Additional Comments On Reviewer Discussion:**

All reviewers give the recommondation of rejection and the authors do not provide any response during and after rebuttal period. Thus, this paper should be rejected.

---

### Decision · Program_Chairs · 2025-01-22

Reject